# Preventive Healthcare and Management for Acute Lymphoblastic Leukaemia in Adults: Case Report and Literature Review

**DOI:** 10.3390/healthcare9050531

**Published:** 2021-05-02

**Authors:** Wei-Ping Chen, Wen-Fang Chiang, Hung-Ming Chen, Jenq-Shyong Chan, Po-Jen Hsiao

**Affiliations:** 1Department of Internal Medicine, Taoyuan Armed Forces General Hospital, Taoyuan 325, Taiwan; windwindchen@gmail.com; 2Division of Infectious Disease and Tropical Medicine, Department of Internal Medicine, Tri-Service General Hospital, National Defense Medical Center, Taipei 114, Taiwan; 3Division of Nephrology, Department of Internal Medicine, Taoyuan Armed Forces General Hospital, Taoyuan 325, Taiwan; wfc96076@aftygh.gov.tw; 4Division of Nephrology, Department of Internal Medicine, Tri-Service General Hospital, National Defense Medical Center, Taipei 114, Taiwan; 5Division of Hematology and Oncology, Department of Internal Medicine, Taoyuan Armed Forces General Hospital, Taoyuan 325, Taiwan; mptt@aftygh.gov.tw; 6Department of Life Sciences, National Central University, Taoyuan 320, Taiwan

**Keywords:** Adult acute lymphoblastic leukaemia, risk factors, prognosis, treatment, hypercalcaemia

## Abstract

Acute lymphoblastic leukaemia (ALL) is diagnosed by the presence of at least 20% lymphoblasts in the bone marrow. ALL may be aggressive and include the lymph nodes, liver, spleen, central nervous system (CNS), and other organs. Without early recognition and timely treatment, ALL will progress quickly and may have poor prognosis in clinical scenarios. ALL is a rare type of leukaemia in adults but is the most common type in children. Precipitating factors such as environmental radiation or chemical exposure, viral infection, and genetic factors can be associated with ALL. We report a rare case of ALL with symptomatic hypercalcaemia in an adult woman. The patient presented with general weakness, poor appetite, bilateral lower limbs oedema, consciousness disturbance, and lower back pain for 3 weeks. She had a history of cervical cancer and had undergone total hysterectomy, chemotherapy, and radiation therapy. Her serum calcium level was markedly increased, at 14.1 mg/dl at admission. Neck magnetic resonance imaging, abdominal sonography, abdominal computed tomography, and bone marrow examination were performed. Laboratory data, including intact parathyroid hormone (i-PTH), peripheral blood smear, and 25-(OH) D3, were checked. Bone marrow biopsy showed B cell lymphoblastic leukaemia. Chemotherapy was initiated to be administered but was discontinued due to severe sepsis. Finally, the patient died due to septic shock. This was a rare case of B cell ALL in an adult complicated by hypercalcaemic crisis, which could be a life-threatening emergency in clinical practice. Physicians should pay attention to the associated risk factors. Early recognition and appropriate treatment may improve clinical outcomes.

## 1. Introduction

Acute lymphoblastic leukaemia (ALL) is a malignancy in which lymphoid precursors replace the normal bone marrow. According to the National Comprehensive Cancer Network guidelines, the diagnosis of ALL requires 20% or more blast cells in bone marrow aspirate and biopsy materials. Peripheral blood may be a substitute for bone marrow for diagnosis. More than 1000 circulating lymphoblasts per microlitre of peripheral blood are required. There are multiple symptoms of acute lymphoblastic leukaemia. Fever is the most common sign of ALL. Symptoms of anaemia include fatigue, dizziness, and dyspnoea. Bleeding manifestations, lymphadenopathy, coagulopathy, increased susceptibility to infections, and hepatosplenomegaly are common. Hypercalcaemia (the presence of a corrected serum calcium level above 10.5 mg/dl) can be also a rare clinical presentation of ALL. Hypercalcaemic crisis (the presence of a corrected serum calcium level above 14 mg/dl) is a life-threatening emergency. Aggressive screening for hypercalcaemia is important. Delayed diagnosis may lead to a worse clinical outcome. In patients with consciousness disturbance, hypercalcaemia, kidney injury, and bone pain, the possibility of malignancy should be considered. Aggressive screening for hypercalcaemia is important because some malignancies may share common features [1]. Elevated serum parathyroid hormone-related peptide (PTHrP), TNF-α, IL-6, and soluble IL-2 receptors are associated with hypercalcaemia in patients with ALL [1,2,3,4,5].

Most cases of ALL complicated hypercalcaemia occur in children [1,2,3,4,5]. Here, we presented an adult case of ALL with hypercalcaemic crisis. The final diagnosis was made via bone marrow examination. ALL is the most prevalent malignancy in children. A good outcome can be achieved after the appropriate treatments. Nonetheless, the prognosis of ALL in adults is not usually optimistic. Current research about the environmental risk factors for ALL in adults is still scant. In this report, we investigated the risk factors, clinical manifestations, diagnosis, treatments, and outcomes of ALL in adults.

## 2. Case Presentation

A 67-year-old female was admitted to the Internal Medicine ward of Taoyuan Armed Forces General Hospital in August 2018 in Taiwan because she had suffered from symptoms of general weakness, generalized tenderness, poor appetite, slurred speech, and lower back pain for 3 weeks. Her family history was unremarkable. The woman had a history of cervical cancer and had undergone total hysterectomy, chemotherapy, and radiation therapy. Her body temperature was 36.6 °C, her blood pressure was 101/57 mmHg, her pulse rate was 109 beats/min, and her respiratory rate was 18 breaths/min. Physical examination demonstrated myoclonic jerk movements, bilateral lower limb oedema, and consciousness disturbance. Blood laboratory investigation revealed a white blood cell count of 8810/µL (normal range, 4800–10,800), a haemoglobin level of 11.6 g/dl (normal range, 12–16), low platelet counts of 57,000/µL (normal range, 130,000–400,000), serum calcium level of 14.1 mg/dl (normal range, 8.5–10.5), a serum albumin level of 3.95 g/dL (normal range, 3.5–5.5), an intact parathyroid hormone (i-PTH) level of 5.69 pg/mL (normal range, 15–65), 25-(OH)D3 level of 9.6 ng/mL (normal range, 30–100), giant platelets, nucleated red blood cells (which were seen in a peripheral blood smear), and a phosphate level of 4.71 mg/dl (normal range, 2.4–4.1). Serial laboratory data are shown in Table 1. The patient was transferred to the intensive care unit for monitoring. Neck magnetic resonance imaging showed enlarged lymph nodes over the right level IIa of the neck. Abdominal sonography revealed moderate fatty liver disease. Abdominal contrast enhanced computed tomography (CT) revealed multiple small calcifications scattered in the pancreas. Serum protein electrophoresis was performed but showed no specific findings. Bone marrow biopsy, flow cytometry, and the pathology report (Figure 1a,b) showed acute B cell lymphoblastic leukaemia (B-ALL) negative for a BCR-ABL translocation. The blast population lacked cytoplasmic immunoglobulin. Flow cytometry identified deoxynucleotide transferase (TdT)-, CD10-, CD19-, CD22-, CD25-, and CD34-positive cells. The blast population showed the expression of HLA-DR, CD19, CD10, CD22, and CD34, consistent with pre-B-cell acute lymphoblastic leukaemia. Chemotherapy with HYPER-CVAD was administered 21 days after admission. However, chemotherapy was discontinued due to pancreatitis and urinary tract infection. The patient complained of abdominal pain for several weeks during hospitalization, which may have been due to chronic pancreatitis. Medication with adequate hydration, loop diuretics, and calcitonin were administered for hypercalcaemia. Urinary tract infection and hospital-acquired pneumonia developed. Her sputum culture yielded *Carbapenem*-resistant *Acinetobacter baumannii*, and urine culture yielded extensively drug-resistant *Proteus mirabilis* and *Escherichia coli*. The patient died due to septic shock.

## 3. Discussion

### 3.1. Anatomy and Risk Factors for ALL

ALL occurs more frequently in children [1], and the evidence of predisposing factors is more common in children. ALL arises from the uncontrolled proliferation of lymphoid precursor cells. According to the American Cancer Society, the incidence of ALL peaks at ages younger than 5 years and declines until the mid-20s. The risk then begins to rise again at ages older than 50 years. ALL may occur at any age in life. This disease develops due to genetic alterations that suppress the normal process of haematopoiesis. B- and T-cell lymphoblasts present leukaemia-associated antigens that correspond to their own lineage development. Precursor B-cell ALL cells are typically HLA-DR^+^, CD10, CD19, TdT, and/or CD79a^+^, and/or CD22^+^, and/or CD34 on the surface [6]. Precursor T-cell ALL cells usually express CD2, CD7, and cytoplasmic CD3^+^ and are positive for one or more of the myeloid/stem cell markers CD34, CD117, HLADR, CD13, CD33, CD11b, or CD65 [7]. Current evidence related to ALL in children includes genetic susceptibility, environmental factors (such as pesticides, radiation, and benzene), birth weight, delivery by Caesarean section [8], maternal infection during pregnancy [9], maternal exposure to X-rays during pregnancy [10], maternal body weight [11], maternal diabetes, paternal tobacco use, exposure during preconception, pregnancy, and the postnatal period in children aged more than 15 years [12]. Most studies associated with the risk of ALL have focused on children. To date, established risk factors including tobacco use, obesity, family history, environmental exposure, alcohol consumption, genetic susceptibility, genotoxic chemotherapy, and exposure to radiation have been reported (Table 2). One previous study showed an association between AML, ALL, and tobacco. The risk was greater among subjects who had smoked for at least 10 years [13]. Tobacco smoke has been linked to chromosomal meiotic errors [14]. One cohort study showed no statistically significant association between tobacco and ALL [15]. Polymorphisms of the MDM2 and P53 genes increased the risk of adult ALL [16]. A family history of a haematological malignancy, working in the leather industry, working in the textile industry, alcohol consumption, and starting to use alcohol before 21 years of age were associated with an increased risk of adult ALL [17]. One meta-analysis of cohort studies revealed that obesity in men was positively associated with ALL [18]. One study reported that body mass index was not significantly associated with ALL in women [19]. Another meta-analysis of the prospective cohort study also showed that body mass index was not associated with the incidence of ALL in women [20]. Further studies focusing on this effect are lacking. According to a research article, obese and control BCR/ABL mice had a similar median survival. Older high-fat diet-induced obese mice had accelerated ALL onset. Obese AKR mice had increased levels of insulin, leptin, and IL-6 [21]. Insulin has been shown to increase ALL cell proliferation in vivo [22]. Leptin has been shown to stimulate haematopoietic progenitors and multiple leukaemia cell types [23]. Prior exposure to chemotherapy or radiotherapy has been postulated to be a risk factor for leukaemia in a patient [24]. Patients who had previous malignancy and received chemotherapy or radiation therapy before the diagnosis of ALL are considered to have therapy-related ALL. This subgroup had poor cytogenetic and molecular features, lower response rates to standard therapies, and inferior survival outcomes [7].

### 3.2. Molecular Genetics and Clinical Manifestations for ALL

The Philadelphia (Ph) chromosome is caused by a translocation of chromosomes 9 and 22. The BCR/ABL fusion gene is derived from this translocation. It is found in 95% of chronic myeloid leukaemia cases, in 17–25% of acute lymphoblastic leukaemia cases, and in 0.9 to 3% of acute myeloid leukaemia (AML) cases [25]. Some patients have molecular evidence of the BCR-ABL fusion gene without the Ph chromosome being detected by conventional cytogenetics. The BCR-ABL fusion gene may be detectable only by fluorescence in situ hybridization (FISH) or reverse-transcriptase polymerase chain reaction (RT-PCR) [26,27]. For patients with ALL with unsuccessful cytogenetic analysis or normal karyotypes under conventional cytogenetic analysis, it is possible to determine aberrations with comparative genomic hybridization [28]. Several methods have been developed to understand the leukaemogenesis of ALL. These methods, including cytogenetic analysis with G-bending, 24-colour FISH hybridization, interphase FISH, and polymerase chain reaction, help to identify different subgroups of ALL. Different subgroups have different prognoses [28,29]. Current well-established molecular subtypes with poor/intermediate prognoses are those involving BCR-ABL1 fusion, KMT2A fusions, TCF3-PBX1, and HLF fusions, and low hypodiploid. Double homeobox 4 gene (DUX4), myocyte enhancer factor 2D (MEF2D), and zinc finger protein 384 (ZNF384) are associated with DNA binding factor rearrangements [30]. Ph-like ALL constitutes several genetic alterations, including rearrangements of CRLF2, ABL-class genes, mutations or deletions activating the JAK-STAT or MAPK signaling pathways, other rare kinase alterations, JAK2, and EPOR [31]. PAX5-driven subtypes include PAX5 alterations and hotspot mutations. Rearrangements of BCL2/MYC and/or BCL6 are rare in ALL. IKZF1 is a significant transcription factor that affects the differentiation of B-cell precursors. The IKZF1 p.Asn159Tyr (N159Y) point mutation is rare in B-cell precursor acute lymphoblastic leukaemia [30]. Because molecular genetics is not available in our hospital, information on the genetic alterations of our patient was not identified.

ALL accounts for 28% of all newly diagnosed cases of cancer in children and 1% of all newly diagnosed cases of cancer in adults [32]. Fever, joint pain, osteolytic bone lesions [33,34,35,36,37,38], bleeding manifestations, lymphadenopathy [39], frequent infection, coagulopathy, and hepatosplenomegaly are common [40]. Some individuals presented with abdominal pain, consciousness disturbance, nausea, weakness, constipation, numbness, hypereosinophilia [41], and atypical lymphoid cells in peripheral blood smears [42]. Hypercalcaemic crisis is a rare manifestation of ALL [43]. Although rare, ALL can also cause hypercalcaemia.

### 3.3. Diagnosis for ALL

Haematologic derangements in ALL include anaemia, neutropenia, thrombocytopenia, and leukocytosis. Workup should include patient medical history, physical examination, and laboratory data (complete blood count, platelet count, differential counts, uric acid, blood urea nitrogen, creatinine, potassium, phosphate, calcium, bilirubin, and hepatic transaminases and screening for disseminated intravascular coagulation (DIC)) [44]. A bone marrow biopsy and aspirate are required for morphologic assessments, comprehensive flow cytometric immunophenotyping, molecular characterization, and the karyotyping of G-banded metaphase chromosomes [44]. Through the newly developed methods for diagnosis and subgrouping, we can refine risk stratification and select the optimal treatment protocol. The differential diagnosis of ALL usually includes AML, hairy cell leukaemia, and malignant lymphoma. Appropriate and accurate diagnosis is imperative due to the difference in prognoses and treatments.

### 3.4. Hypercalcaemia with ALL

The most common cause of hypercalcaemia in patients who present to the emergency department is malignancy, which accounts for 35% of cases [45]. The mechanism of hypercalcaemia varies with the type of malignancy. In clinical practice, the common aetiology of hypercalcaemia includes drugs, excessive vitamin D intake, endocrine disorder, factitious hypercalcaemia, malignancies with lytic bone lesions or PTHrP hypersecretion, excessive 1,25(OH)_2_D3 production, hyperparathyroidism, etc [46].

We reviewed the literature about the clinical manifestations of ALL in adults and investigated the mechanism of hypercalcaemia (Table 3). Shimonodan found that PTHrP produced from lymphoblasts was associated with relapse and poor prognosis in ALL [47]. In 2001, Fukasawa, Kato, Fujigaki, Yonemura, Furuya and Hishida [48] noted that TNF-α, IL-6, and the soluble IL-2 receptor were elevated in patients with B-ALL. Multiple myeloma, leukaemia, lymphoma, and non-haematologic malignancies share the same features as hypercalcaemia. Breast cancer, myeloma, and lymphoma are characterized by lytic skeletal metastasis. Breast cancer is the most common malignancy associated with hypercalcaemia [49]. PTHrP hypersecretion is associated with squamous cell carcinomas, breast carcinomas, renal cortical carcinomas, adult T cell leukaemia syndrome [50,51], and myeloid and lymphoid haematological malignancies [52]. Careful differential diagnosis is important. Elevated i-PTH is likely caused by primary hyperparathyroidism. Hypercalcaemia due to primary hyperparathyroidism usually presents no symptoms [53]. On the other hand, primary hyperparathyroidism has a relatively benign course and is characterized by high calcium and i-PTH levels with low phosphorous levels. If the i-PTH level is in the mid-upper normal range or is minimally elevated, familial hypocalciuric hypercalcaemia and primary hyperparathyroidism should be taken into consideration [54]. Familial hypocalciuric hypercalcaemia presents upper normal or mildly elevated i-PTH levels with high magnesium and an exceptionally low urinary calcium-to-creatinine ratio. Patients with hypercalcaemia of malignant aetiology are usually more symptomatic [53]. If the i-PTH level is low, PTHrP should be checked. If PTHrP is elevated, humoural hypercalcaemia of malignancy is more likely. Further imaging, bone marrow biopsy, tumour screening, CD markers, light protein chains of kappa and lambda should be checked.

The highest incidence of hypercalcaemia in haematological malignancy occurs in myeloma. Hypercalcaemia was present in 32% of myeloma patients [46]. Hypercalcaemia occurs in 15% of patients with lymphoma during the course of the disease and rarely occurs at presentation. B cell lymphoma with hypercalcaemia is less frequent (7–8%) than other types of lymphoma [55]. According to a retrospective study of 1,200 patients with B-CLL, only 7 patients (0.006%) had hypercalcaemia [56]. According to a study that included 83 patients with B-ALL, the incidence of hypercalcaemia was noted to be 4.8% in children [57]. Statistical data on adult ALL with hypercalcaemia has not been established. Hypercalcaemia can influence multiple systems. In the cardiovascular system, hypercalcaemia causes bradycardia, hypertension, bundle branch/AV block, and cardiac arrest. Neuropsychiatric disturbance includes anxiety, depression, and cognitive dysfunction. More severe symptoms are more likely to occur in older adults and in those who have rapid increases in calcium concentrations. Severe central nervous system (CNS) and neurologic symptoms include lethargy, confusion, stupor, and coma [58]. Polyuria, polydipsia, nocturia, hypercalciuria, nephrolithiasis, nephrocalcinosis, and renal failure are complications of hypercalcaemia in the renal system. Gastrointestinal symptoms include nausea, vomiting, anorexia, abdominal pain, peptic ulcers, and pancreatitis. Bone pain, arthralgia, osteopenia, and osteoporosis were associated with lytic bone lesions [42].

Treatments for hypercalcaemia can be initiated based on the level of calcium and the symptoms. Adequate hydration is recommended for mild hypercalcaemia (< 12 mg/dl). Moderate hypercalcaemia (12–14 mg/dl) without obvious symptoms may not require immediate therapy. Treatment for hypercalcaemia is indicated if there are symptoms of hypercalcaemia or if the serum calcium level is more than 14 mg/dl, regardless of symptoms. Factors that may aggravate hypercalcaemia should be avoided. The treatment goals include increasing renal calcium excretion, reducing gastrointestinal absorption, and the slowing of bone resorption. Immediate therapy is directed at volume expansion with isotonic saline to achieve a urine output of ≥ 200 mL/h to 300 mL/h. Loop diuretics should be used only when the intravascular volume has been restored. Intestinal calcium absorption can be reduced by avoiding calcium-containing foods and vitamin D supplements. When bone resorption is the main source of calcium, calcitonin and bisphosphonates are the treatments of choice. In patients with congestive heart failure or renal insufficiency, haemodialysis therapy should be considered for a rapid reduction in serum calcium levels. For patients with bisphosphonates contraindication, denosumab with calcitonin may be administered concurrently. Calcimimetics can be used in patients with severe hypercalcaemia due to parathyroid carcinoma [59] and in patients on haemodialysis with secondary or tertiary hyperparathyroidism [60].

### 3.5. Treatments for ALL

Current chemotherapy for ALL includes four phases: induction, consolidation, intensification, and long-term maintenance. Induction therapy includes glucocorticoids, vincristine, L-asparaginase, anthracycline, and intrathecal chemotherapy. Consolidation usually consists of glucocorticoids, vincristine, cytarabine, high-dose methotrexate, asparaginase, and mercaptopurine over a 12-week period. Maintenance therapy is composed of mercaptopurine and weekly methotrexate [61]. In some cases, vincristine and prednisone may be combined. The folate analog methotrexate (MTX) and the purine analog antimetabolite 6-mercaptopurine (6MP) became pioneering anticancer agents as maintenance and consolidation therapy.

Genetic analysis is used to classify the ALL subtype, risk stratify the patients, and select the appropriate therapy. Patients with BCR-ABL1 translocations had a poor prognosis, but tyrosine kinase inhibitors improved the outcome Imatinib combined with chemotherapy led to a high complete remission rate (97%) [62]. Allogenic haemopoietic cell transplantation remains the first-line consolidation therapy for the Ph-positive subtype and high-risk patients who have an available donor [61]. MEF2D-fusion (M-fusion) genes are related to survival-supporting ability of leukemic cells. Venetoclax activates caspase and induce proteolysis of M-fusion proteins [63].

Mitochondria dysfunctions were widely investigated recently as a target for anticancer handling because they could be involved in a metabolic reprogramming of a cancer cell and might play an important role in reactive oxygen species (ROS) generation, Ca^2+^ signaling, and cell death. Consequently, voltage-dependent anion channel 1 (VDAC1)-directed drugs can further encourage the suppression of aerobic glycolysis, closure of VDAC-1, overloading of mitochondrial Ca^2+^, stoppage of the oxidative phosphorylation, and further inhibition of oxidative stress [64]. Phenolic compounds killed cancer cells through 2 mechanisms: a direct uncoupling effect and mitochondrial Ca^2+^ overload [65]. However, studies of these compounds for ALL lacked. Recently, immunotherapies have been developed. Immunotherapies include monoclonal antibodies, antibody–drug conjugates, and chimeric antigen receptor (CAR) T cells. Inotuzumab ozogamicin is an antibody–drug conjugate and has been indicated for use in adults and children with refractory or relapsed B-ALL [66]. Chimeric antigen receptor (CAR) T cells are genetically engineered to express a receptor that binds to a tumour antigen [67]. CAR T cells were approved for relapsed or refractory pediatrics and young adults with precursor B-ALL [68,69]. ABL001 (Asciminib) is an investigational treatment that binds to the myristoyl site of the BCR-ABL1 protein [70]. Due to cytogenetic studies for ALL develop, the treatments are advancing rapidly. Clinical trials hope to test the effectiveness and safety of these drugs.

Pancytopenia is a common side effect of cyclophosphamide. Our patient received cyclophosphamide initially but discontinued its use due to severe sepsis, chronic pancreatitis exacerbation, and pancytopenia. Cyclophosphamide has been listed as a rare cause of drug-induced pancreatitis [71]. In addition to hypercalcemia, which may be associated with a risk of pancreatitis, cyclophosphamide was considered the possible cause of pancreatitis in our patient. A recent retrospective case-control study demonstrated that approximately 90% of patients with ALL developed hypercalciuria during chemotherapy. In addition, hypercalciuria was highly associated with corticosteroid administration among these patients [72]. Adequate hydration is important in patients with ALL receiving steroid-containing chemotherapy to prevent urolithiasis.

The side effects of anti-cancer drugs are a major public health problem. For example, the response to treatment drugs is subject to wide inter-individual variability, for which toxicity is a crucial issue. Pharmacogenetics (Pgx) is a rapidly growing field studying how genetic differences influence the variability of individual patient responses to drugs. Pharmacogenetics aims to personalize the treatment of ALL. Therapy-related toxicity causes the discontinuation of chemotherapy and influences the quality of life of the patients and the risk of recurrence. In one comprehensive pharmacogenetic study of leukaemia, several inherited polymorphisms were significantly related to the toxicity of anti-leukaemic therapy [61]. The strongest predictor of hyperbilirubinemia was the UGT1A1 promoter–repeat polymorphism [61]. The RFC-1 genotype is associated with hepatotoxicity related to methotrexate in the consolidation phase [73]. Genetic polymorphisms in the cytochrome P450 (CYP), glutathione S-transferase (GST), and aldehyde dehydrogenase (ALDH) families of enzymes have been found to impact the response to and/or toxicity associated with cyclophosphamide-based therapies [74].

### 3.6. Outcomes and Associated Genetic Susceptibility in ALL

In addition to environmental risk factors, genetic susceptibility has been identified. Adult patients with ALL are at risk of developing CNS involvement during their disease. This is particularly true for patients with L3 (Burkitt) morphology. The subtype of acute lymphoblastic leukaemia based on chromosomal abnormalities is associated with prognosis and risk stratification. B-ALL subtypes implicate chromosomal rearrangements and aneuploidy [61]. Chromosome translocations include the rearrangement of mixed lineage and fusion genes. Mixed-lineage leukaemia is associated with a poor prognosis [75]. ETV6-RUNX1 (t (12;21) (p13; q22)) and TCF3-PBX1 (t (1;19) (q23; p13)) are associated with a favourable prognosis [76]. The frequency of BCR-ABL (Ph-chromosome) (t (9;22) (q34; q11)) in adults is more than 25%. This gene fusion event is associated with a poor prognosis, but the prognosis improved after treatment with tyrosine kinase inhibitors [77,78,79]. Genomic alterations in Ph-like leukaemia affect transcription factors, cytokine receptors, and tyrosine kinase signaling [61]. Myocyte enhancer factor 2D (MEF2D)-rearranged B-ALL is a subtype with an older onset with enhanced MEF2D transcriptional activity that is associated with poor outcomes [80]. DUX4, ZNF384 and MEF2D fusion genes account for 40% of Ph-negative B cell adolescent and young adult-ALL DUX4 and ZNF384 fusions were associated with longer disease-free survival after complete remission than Ph-like ALL [81]. PAX5 is a tumour suppression gene. PAX5 translocations inhibit the activity of PAX5 and promote the development of B cell precursor leukaemia [82]. T cell-acute lymphoblastic leukaemia (T-ALL) is an aggressive haematologic malignancy that accounts for approximately 20% of all cases of ALL. T-ALL tends to be more common in adults than in children [83]. The activation of NOTCH [84], mutation of FBXW7 [85], overexpression of oncogenic transcription factors (TAL1, TAL2, LYL1, OLIG2, LMO1, LMO2, TLX1, TLX3, NKX2-1, NKX2-2, NKX2-5, HOXA genes, MYC, MYB, and TAN1) [86], CDKN2B downregulation [82], and alterations in the signal transduction pathway (PTEN) were present in T cell ALL. Another clinical feature of hypercalcaemia can be associated with t (17;19) in patients with ALL; this has been reported based on the effect of PTHrP [87].

## 4. Conclusions

In conclusion, ALL is a rare type of leukaemia in adults and may have various clinical manifestations. Some factors (such as age, CNS involvement, cellular morphology, and chromosomal abnormalities) could be associated with a clinical prognosis. Hypercalcaemic crisis is also a rare manifestation of ALL and may have a poor clinical outcome. Aggressive screening for hypercalcaemia is important because the underlying cause may be related to malignancy. If a diagnosis is made earlier and appropriate treatment is provided, hypercalcaemia can be controlled. Surveying associated risks is also imperative to provide the preventive healthcare required for the development of ALL.

## Figures and Tables

**Figure 1 healthcare-09-00531-f001:**
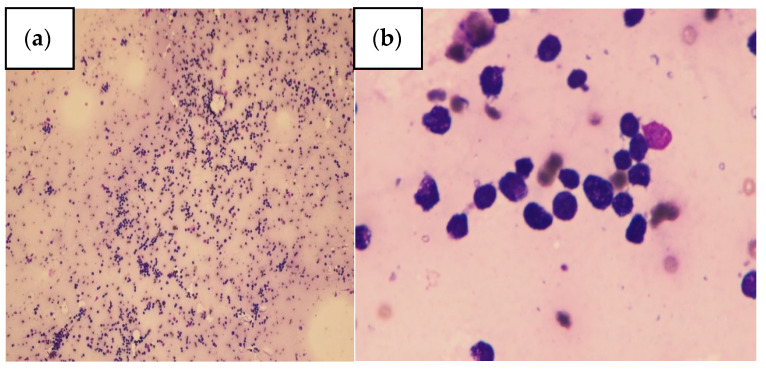
Bone marrow pathology. (**a**) Bone marrow cytology showed hypercellular marrow for age (90%), hypoplasia of myeloid, erythroid series and megakaryocytes; (**b**) Bone marrow cytology revealed bone marrow filled with lymphoblasts (86.1%).

**Table 1 healthcare-09-00531-t001:** Blood biochemistry data.

Parameters	Results	Normal Value
White blood cell count (/µL)	8810	4800–10,800
Haemoglobin (g/dL)	11.6	12–16
Platelet count (/µL)	57,000	130,000–400,000
Mean corpuscular volume (fL)	83.6	81–99
BUN (mg/dL)	43.7	15–40
Creatinine (mg/dL)	1.21	0.9–1.8
Sodium (mEq/L)	131	133–145
Potassium (mEq/L)	4.74	3.8–5.0
Chloride (mEq/L)	92.4	96–106
Calcium (mg/dL)	14.1	8.5–10.5
Phosphate (mg/dL)	4.71	2.4–4.1
Uric acid (mg/dL)	21	1.9–7.5
GOT (U/L)	61.2	10–40
GPT (U/L)	36.9	7–56
Globulin (gm/dL)	2.75	1.4–3.5
Albumin (gm/dL)	3.95	3.5–5.5
A/G ratio	1.4	0.8–2.0
25-(OH) D3 (ng/mL)	9.6	30–100
i-PTH (pg/mL)	5.69	15–65

Abbreviations: BUN: blood urea nitrogen; GPT: glutamyl pyruvate transaminase; GOT: glutamyl oxaloacetic transaminase; A/G: albumin/globulin; i-PTH: intact parathyroid hormone; 25-(OH) D3: 25-hydroxy vitamin D.

**Table 2 healthcare-09-00531-t002:** Risk factors for ALL in adults.

Report	Country	Period/Age	Case Number	Risk factors	Odds Ratio and Relative Risk	95% CI
Kane, Roman, Cartwright, Parker and Morgan [13]	Italy	1991–1996/16–69	100	Smoked at least once a day and for at least 6 months	Years of smoking/odds ratio:	
					10–19 years/2.1	0.9–4.7
					20–29 years/1.0	0.4–2.6
					30–39 years/1.0	0.4–2.8
					>40 years/10.6	1.2–90.5
Skibola, Slager, Berndt, Lightfoot, Sampson, Morton and Weisenburger [17]	Europe, North America, and Australia	NA/18–91	152		Odds ratio	
				First-degree had a haematologic malignancy	2.6	1.22–5.54
				Leather worker	3.91	1.35–11.35
				Sewer and embroiderer	4.38	1.41–13.62
				Former alcohol consumption	5.87	1.74–19.77
				Current alcohol consumption	2.48	0.99–6.19
Psaltopoulou, Sergentanis, Ntanasis-Stathopoulos, Tzanninis, Riza and Dimopoulos [18]	NA	NA	4 men	Obesity	Relative risk 1.69	1.04–2.73
Engeland, Tretli, Hansen and Bjorge [19]	Norway	1963–2001/20–74	119 men	Obesity	Relative risk 2.77	1.49–5.12
Castillo, Reagan, Ingham, Furman, Dalia, Merhi, Nemr, Zarrabi and Mitri [20]	NA	NA	NA	Obesity	Relative risk 1.62	1.12–2.32
Tang, Zuo, Thomas, Lin, Liu, Hu, Kantarjian, Bueso-Ramos, Medeiros and Wang [24]	America	2004–2010	457	Alkylating agents or topoisomerase II inhibitors	Intervals from prior malignancy to the onset of precursor B-ALL in patients with secondary precursor B-ALL were significantly shorter in the cytotoxic therapies group: 36 months versus 144 months (*p* < 0.001)	

Abbreviation: CI: confidence interval; NA: not available; B-ALL: B cell acute lymphoblastic leukaemia.

**Table 3 healthcare-09-00531-t003:** Hypercalcaemia in adult patients with ALL: a review and comparison of the literature.

Reports	Age/Sex	Ca mg/dl	Chromosomal Testing and Immunohistochemical Analysis	Clinical Manifestations	Mechanism of Hypercalcaemia	Survival Time
Granacher, Berneman, Schroyens, Van de Velde, Verlinden and Gadisseur [36]	34/male	12.8	CD10, CD19, CD34, CD33, TdT CD79a, t (9,22) (q34, q11,2)	Vertebrae and rib osteolytic bone lesions	NA	CR
Kaiafa, Perifanis, Kakaletsis, Chalvatzi and Hatzitolios [33]	24/male	13.3	CD19, CD10, iCD22, TdT, iCD79a, CD34, CD38, HLA-DR, CD11b, CD13, iMPO, 46, XY, dup (1) (q21q32), del (8) (p22) [12]/46, XY [8]	Osteolytic lesions in all lumbar vertebrae, the sacrum, both femora and the ilium	Induced renal failure	2 years
Zou, Shen, Zhu, Zhang and Zhu [34]	47/male	17.8	CD34, CD10, CD20, bcl-2	Abdominal pain, vomiting, bone pain, anaemia, neutropenia, and renal insufficiency	NA	NA
Mahmood, Ubaid and Taliya Rizvi [1]	22/male	14.6	TdT, CD 10, CD 79a	Pain and generalized weakness, mild anaemia, osteolytic lesions in the iliac bones and cranium	NA	NA
Chung, Kim, Choi, Yoo and Cha [37]	35/male	18.2	NA	Osteolytic lesion of the mandible. Dull pain on the right posterior mandible. Left third and sixth nerve palsy	PTHrP (1.5 pmol/l)	7 days, died from pneumonia, multiple organ failure and shock.
Fukasawa, Kato, Fujigaki, Yonemura, Furuya and Hishida [48]	53/female	15.2	CD10, CD19, HLA-DR	Drowsiness, nausea, bone pain, multiple osteolytic lesions in the skull and ribs	TNF-α, IL-6, and soluble IL-2 receptor were increased	In complete remission after 26 months of chemotherapy
Zhou, Tang, Liu and Li [35]	53/female	15.5	CD19, CD22, CD34	Nausea and vomiting, compression fracture and degeneration of the lumbar vertebra and skull	NA	NA
Stein and Boughton [38]	50/male	12.5	CD19, CD24, CD10	Multiple small lytic lesions of the skull and severe osteoporosis of the spine with partial collapse of two thoracic vertebrae	NA	NA
Foss, Aquino and Ferry [39]	72/male	13.1	CD3, CD4, CD2, T cell receptorα/β, CD45RO	Abdominal pain, liver and renal dysfunction, respiratory insufficiency, changes in mental status	NA	NA
Fathi, Chen, Carter and Ryan [51]	38/male	17.7	CD2, CD3, CD25	Circulating abnormal T cells, cervical and inguinal lymphadenopathy, splenomegaly, nausea, abdominal pain, fatigue, nonbilious and nonbloody emesis, right pulmonary embolism, hypercalcaemia	Increased osteoclast activity	Died at home 9 months after initial diagnosis

Abbreviation: Ca: calcium; CD: cluster of differentiation; CR: complete remission; NA: not available.

## Data Availability

The data underlying this article will be shared upon reasonable request to the corresponding author.

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
