# Peer review of "Preventive Healthcare and Management for Acute Lymphoblastic Leukaemia in Adults: Case Report and Literature Review"

_healthcare, 2021, doi:10.3390/healthcare9050531_

Round 1
Reviewer 1 Report
Minor Comments
Hypercalcemia is a potentially fatal disorder and its role in adult ALL is now well dissected. This case study mainly highlighted the adverse effect of hypercalcaemic crisis in adult ALL and suggested that appropriate and timely diagnosis and treatment strategies are crucial for the proper management of these patients. The topic is of interest for both researchers and clinicians. The case report is very comprehensive and adequately covers the current literature in the field.
I got two recommendations that could make the case study more appealing to the readership.
- This case study emphasize the adverse effect of hypercalcemia in adult ALL without offering a discussion or opinion for the differential diagnosis and effective management strategies. It is worth to refer and discuss these information in the manuscript.
- A strong association between ALL patient’s chemotherapy and hypercalciuria has been shown (Hori et al. Pediatr Int. 2020 Oct 30). Additionally, impact of mitochondrial activity in ALL relapse and Calcium signaling is demonstrated. The authors should include these informations in the discussion.
Author Response
Reviewer 1
Hypercalcemia is a potentially fatal disorder and its role in adult ALL is now well dissected. This case study mainly highlighted the adverse effect of hypercalcaemic crisis in adult ALL and suggested that appropriate and timely diagnosis and treatment strategies are crucial for the proper management of these patients. The topic is of interest for both researchers and clinicians. The case report is very comprehensive and adequately covers the current literature in the field. I got two recommendations that could make the case study more appealing to the readership.
Author Reply: We are deeply honored by the time and effort you spent in reviewing this manuscript. We have revised the manuscript thoroughly according to your suggestions. The responses to your comments are below.
- This case study emphasizes the adverse effect of hypercalcemia in adult acute lymphoblastic leukaemia (ALL) without offering a discussion or opinion for the differential diagnosis and effective management strategies. It is worth to refer and discuss this information in the manuscript.
Author Reply: Thanks for your suggestion. We have amended the manuscript as the following as well.
Careful differential diagnosis is important. Elevated i-PTH is likely caused by primary hyperparathyroidism. Hypercalcaemia due to primary hyperparathyroidism usually has no symptoms [53]. On the other hand, primary hyperparathyroidism has a relatively benign course and is characterized by high calcium and i-PTH levels with low phosphorous levels. If the i-PTH level is in the mid-upper normal range or is minimally elevated, familial hypocalciuric hypercalcaemia and primary hyperparathyroidism should be taken into consideration [54]. Familial hypocalciuric hypercalcaemia has upper normal or mildly elevated i-PTH levels with high magnesium and an exceptionally low urinary calcium-to-creatinine ratio. Patients with hypercalcaemia of malignant aetiology are usually more symptomatic [53]. If the i-PTH level is low, PTHrP should be checked. If PTHrP is elevated, humoural hypercalcaemia of malignancy is more likely. Further imaging, bone marrow biopsy, tumour screening, CD markers, light protein chains of kappa and lambda should be checked.
Treatments for hypercalcaemia can be initiated based on the level of calcium and symptoms. Adequate hydration is recommended for mild hypercalcaemia (< 12 mg/dl). Moderate hypercalcaemia (12-14 mg/dl) without obvious symptoms may not require immediate therapy. Treatment for hypercalcaemia is indicated if there are symptoms of hypercalcaemia or if the serum calcium level is more than 14 mg/dl, regardless of symptoms. Factors that may aggravate hypercalcaemia should be avoided. The treatment goals include increasing renal calcium excretion, reducing gastrointestinal absorption and slowing bone resorption. Immediate therapy is directed at volume expansion with isotonic saline to achieve a urine output of ≥ 200 mL/h to 300 mL/h. Loop diuretics should be used only when the intravascular volume has been restored. Intestinal calcium absorption can be reduced by avoiding of calcium-containing foods and vitamin D supplements. When bone resorption is the main source of calcium, calcitonin and bisphosphonates are the treatment of choice. In patients with congestive heart failure or renal insufficiency, haemodialysis therapy should be considered for a rapid reduction in serum calcium levels. For patients with bisphosphonates contraindication, denosumab with calcitonin may be administered concurrently. Calcimimetics can be used in patients with severe hypercalcaemia due to parathyroid carcinoma [59] and in patients on haemodialysis with secondary or tertiary hyperparathyroidism [60].
- A strong association between ALL patient’s chemotherapy and hypercalciuria has been shown (Hori et al. Pediatr Int. 2020 Oct 30). Additionally, impact of mitochondrial activity in ALL relapse and Calcium signaling is demonstrated. The authors should include these informations in the discussion.
Author Reply:
Thanks for your suggestion. We have amended the manuscript as the following as well in the section of discussion.
A recent retrospective case-control study demonstrated that approximately 90% of patients with ALL developed hypercalciuria during chemotherapy. In addition, hypercalciuria was highly associated with corticosteroid administration among these patients [72]. Adequate hydration is important in patients with ALL receiving steroid-containing chemotherapy to prevent urolithiasis.
Reference
[72] D. Hori, R. Kobayashi, D. Suzuki, K. Kodama, M. Yanagi, S. Matsushima, K. Kobayashi, A survey of hypercalciuria during chemotherapy in acute lymphoblastic leukemia, Pediatrics International n/a(n/a) (2020).
Mitochondria dysfunction were widely investigated recently as a target for anticancer handling because they could be involved in a metabolic reprogramming of a cancer cell and might play an important role in reactive oxygen species (ROS) generation, Ca2+ signaling, and cell death. Consequently, voltage-dependent anion channel 1 (VDAC1)-directed drugs can further encourage the suppression of aerobic glycolysis, closure of VDAC-1, overloading of mitochondrial Ca2+, stoppage of the oxidative phosphorylation, and further inhibition of oxidative stress [64].
Reference
[64] M. Olivas-Aguirre, I. Pottosin, O. Dobrovinskaya, Mitochondria as emerging targets for therapies against T cell acute lymphoblastic leukemia, Journal of leukocyte biology 105(5) (2019) 935-946.

Reviewer 2 Report
The authors described a case of adult ALL presented with hypercalcemia and summarized the current knowledge on disease pathogenesis, complications, therapy.
- In the case presentation they reported the presence of pancreatic calcifications and during disease course the patient developed pancreatitis. Was pancreatitis exacerbation a consequence of the chosen chemo regimen?
- In section 3.2, 3.6 they discussed the molecular genetics of ALL; however in case presentation they did not mention the genetic alteration of the patient.
- Among disease manifestations they discussed hypercalcemia and its treatment, but they did not mention CNS involvement. At least one sentence on this topic should be included.
- The new drugs and new cell therapies for the treatment of high risk adult ALL must be cited among treatment options.
Author Response
Reviewer 2
The authors described a case of adult ALL presented with hypercalcemia and summarized the current knowledge on disease pathogenesis, complications, therapy.
- In the case presentation they reported the presence of pancreatic calcifications and during disease course the patient developed pancreatitis. Was pancreatitis exacerbation a consequence of the chosen chemo regimen?
Author Reply: Thanks for your comments. We have amended the manuscript as the following.
Cyclophosphamide has been listed as a rare cause of drug-induced pancreatitis [71]. In addition to hypercalcemia, which may be associated with a risk of pancreatitis, cyclophosphamide was considered the possible cause of pancreatitis in our patient.
Reference
[71] V.B. Salvador, M. Singh, P. Witek, G. Peress, Cyclophosphamide and doxorubicin-induced acute pancreatitis in a patient with breast cancer, British Journal of Medical Practitioners 7 (2014).
- In section 3.2, 3.6 they discussed the molecular genetics of ALL; however, in case presentation they did not mention the genetic alteration of the patient.
Author Reply: Thanks for your suggestion. We have amended the manuscript as the following.
Because molecular genetics is not available in our hospital, information on the genetic alterations of our patient was not identified.
- Among disease manifestations they discussed hypercalcemia and its treatment, but they did not mention CNS involvement. At least one sentence on this topic should be included.
Author Reply: Thanks for your suggestion. We have amended the manuscript as the following.
Neuropsychiatric disturbance includes anxiety, depression, and cognitive dysfunction. More severe symptoms are more likely to occur in older adults and in those who have rapid increases in calcium concentrations. Severe central nervous system (CNS) and neurologic symptoms include lethargy, confusion, stupor, and coma [58].
- The new drugs and new cell therapies for the treatment of high-risk adult ALL must be cited among treatment options.
Author Reply: Thanks for your invaluable comments. We have amended the manuscript as the following.
Mitochondria dysfunction were widely investigated recently as a target for anticancer handling because they could be involved in a metabolic reprogramming of a cancer cell and might play an important role in reactive oxygen species (ROS) generation, Ca2+ signaling, and cell death. Consequently, voltage-dependent anion channel 1 (VDAC1)-directed drugs can further encourage the suppression of aerobic glycolysis, closure of VDAC1, over-loading of mitochondrial Ca2+, stoppage of the oxidative phosphorylation, and further inhibition of oxidative stress [64]. Phenolic compounds killed cancer cells through 2 mechanisms: a direct uncoupling effect and mitochondrial Ca2+ overload [65]. However, studies of these compounds for acute lymphoblastic leukaemia (ALL) lacked. Recently, immunotherapies have been developed. Immunotherapies include monoclonal antibodies, antibody-drug conjugates, and chimeric antigen receptor (CAR) T cells. Inotuzumab ozogamicin is an antibody-drug conjugate and has been indicated for adults and children with refractory or relapsed B-cell ALL [66]. Chimeric antigen receptor (CAR) T cells are genetically engineered [67] to express a receptor that binds to a tumor antigen. CAR T cells were approved for relapsed or refractory pediatric and young adults with precursor B-cell ALL [68, 69]. ABL001 (Asciminib) is an investigational treatment that binds to myristoyl site of the BCR-ABL1 protein [70].
Reference
[64] S.E. Inzucchi, Understanding hypercalcemia. Its metabolic basis, signs, and symptoms, Postgraduate medicine 115(4) (2004) 69-70, 73-6.
[65] M. Olivas-Aguirre, I. Pottosin, O. Dobrovinskaya, Mitochondria as emerging targets for therapies against T cell acute lymphoblastic leukemia, Journal of leukocyte biology 105(5) (2019) 935-946.
[66] M. Olivas-Aguirre, L. Torres-López, I. Pottosin, O. Dobrovinskaya, Phenolic Compounds Cannabidiol, Curcumin and Quercetin Cause Mitochondrial Dysfunction and Suppress Acute Lymphoblastic Leukemia Cells, International journal of molecular sciences 22(1) (2020).
[67] K.W. Phelan, A.S. Advani, Novel Therapies in Acute Lymphoblastic Leukemia, Current hematologic malignancy reports 13(4) (2018) 289-299.
[68] H.L. Pacenta, T.W. Laetsch, S. John, CD19 CAR T Cells for the Treatment of Pediatric Pre-B Cell Acute Lymphoblastic Leukemia, Paediatric drugs 22(1) (2020) 1-11.
[69] S.L. Maude, T.W. Laetsch, J. Buechner, S. Rives, M. Boyer, H. Bittencourt, P. Bader, M.R. Verneris, H.E. Stefanski, G.D. Myers, M. Qayed, B. De Moerloose, H. Hiramatsu, K. Schlis, K.L. Davis, P.L. Martin, E.R. Nemecek, G.A. Yanik, C. Peters, A. Baruchel, N. Boissel, F. Mechinaud, A. Balduzzi, J. Krueger, C.H. June, B.L. Levine, P. Wood, T. Taran, M. Leung, K.T. Mueller, Y. Zhang, K. Sen, D. Lebwohl, M.A. Pulsipher, S.A. Grupp, Tisagenlecleucel in Children and Young Adults with B-Cell Lymphoblastic Leukemia, The New England journal of medicine 378(5) (2018) 439-448.
[70] A.A. Wylie, J. Schoepfer, W. Jahnke, S.W. Cowan-Jacob, A. Loo, P. Furet, A.L. Marzinzik, X. Pelle, J. Donovan, W. Zhu, S. Buonamici, A.Q. Hassan, F. Lombardo, V. Iyer, M. Palmer, G. Berellini, S. Dodd, S. Thohan, H. Bitter, S. Branford, D.M. Ross, T.P. Hughes, L. Petruzzelli, K.G. Vanasse, M. Warmuth, F. Hofmann, N.J. Keen, W.R. Sellers, The allosteric inhibitor ABL001 enables dual targeting of BCR–ABL1, Nature 543(7647) (2017) 733-737.

Round 2
Reviewer 2 Report
I read the changes. No further comments